# Does the Household Save Water? Evidence from Behavioral Analysis

**Hasrul Hazman Hasan** , **Siti Fatin Mohd Razali \*** and **Nor Hidayah Mohd Razali**

Department of Civil Engineering, Faculty of Engineering & Built Environment, Universiti Kebangsaan Malaysia, UKM Bangi 43600, Selangor, Malaysia; P99749@siswa.ukm.edu.my (H.H.H.); norhidayahmohdrazali96@gmail.com.my (N.H.M.R.)
* Correspondence: fatinrazali@ukm.edu.my; Tel.: +60-38-9216-216

**Abstract:** Management of water supply in urban areas is a challenge that must be faced by water supply companies to ensure the continuity of domestic water supply to the residents in the area. Hence, this study aims to identify local people's behavior and daily activities that led to domestic water wastage. Furthermore, the relationship between the demographic factors of the population trends in reducing water use through water savings in their daily activities or installing a home-saving water system is also undertaken. The data were analyzed and interpreted using IBM SPSS software such as descriptive analysis, covering frequencies, mean and standard deviation, correlation with bivariate correlation, cross-tabulation, and multivariate analysis (MANOVA). Availability and demand in water management will only be managed if water resources and water supply engineers address all the balance sides. It will ensure a more comprehensive and interconnected water sector, ensuring the security and sustainability of water.

**Keywords:** behavioral analysis; water resources; water consumption; domestic water; household; correlation analysis

## 1. Introduction

Water sources are one aspect of environmental sustainability that people need to consider when creating an integrated environmental management system. Climate change and urbanization challenges have made it more relevant to investigate water usage, especially in households. According to the Malaysia National Water Resources Policy, the following principle outlines is the sustainability of water resources for the human well-being, environment, and the development of the country [1]. The uses of clean water resources can be divided into domestic water, which is used by households for daily activities, and non-domestic water is used for commercial, industrial, agricultural, and livestock breeding purposes [2]. The domestic use is higher than other activities as there is an increase in demands for over-population and developed a new challenge for the water management system [3]. Proper domestic water resource management is needed to ensure the supply balance and meet domestic water's current and future consumer demands.

Therefore, the efficient management of water resources demands and supply opportunities through the right water resource management strategies have become increasingly popular nowadays [4]. Various studies were conducted to find solutions to these problems, such as by Corrol et al. [5] finds that the solution to this challenge is a combination of technology and social behavior to promote the conservation of water resources among populations around the world. Metropolitan cities use the most abundant water demands [6]. The study of domestic water consumption by households in a residential area carried out to narrow the scope and clearly see the theoretical influence focusing on the households in the society and thus improve the environmental sustainability, as the use of domestic water for the overall use of water [7].

Water efficiency is essential because water scarcity and uncertainty need to be reconciled with modern society's demands, environmental issues, and affordability of the resource [8]. Water conservation in buildings falls under water demand management, aiming to reduce demand by improving its efficiency, essentially focusing users on more sustainable water consumption approaches. Silva-Afonso [9] described a five-point principle, with a specific focus on water efficiency measures in buildings: (1) Reduce consumption; (2) Reduce loss and waste; (3) Re-use water; (4) Recycle water; and (5) Resort to alternative sources. Previous studies show that the first step to an effective water efficiency initiative is to understand water users' attitudes, preferences, and behavior [10–12].

Households are considered a potential factor in significant water and energy savings [10], and family composition is considered factors that may affect water conservation [13]. Various factors have been identified to affect domestic water use, such as demographic characteristics, socio-demographic, and water supply efficiency. Socio-demographic factors such as household income, type, and size of the dwelling, size of homeownership, family composition, and age may affect daily activities involving water conservation and affect the amount, frequency, and duration of household water consumption [14].

The studies on socio-demographic trends by households in water consumption found that these groups use less than average domestic water, but high water users are among older respondents, families with few children in their households, and families who receive a low annual income [15]. The number of household members affects water consumption, in which households with more family members and larger homes tend to consume domestic water. Hence, this shows that two socio-demographic characteristics that had most influenced domestic water consumption with the total income and larger residence and many occupants [16]. This finding is supported by Frederiks et al. [14], who mention that the life cycle of family and changes in family composition over time was influenced by the level and pattern of household consumption. There was evidence in South East Queensland, where households with small families, the older population's average age, and no children in their average household use lesser domestic water [4].

Gardner and Stern [17] mention two methods to conserve water resources with everyday life behavior and efficient technologies such as low-pressure water pipes and showerheads, high powered washers, and rainwater tanks. The use of water-saving tools is also seen as a water resource management approach toward sustainability. In fact, in 2010, to conserve water resources, the National Water Management Commission (SPAN) introduced new water supply legislation as part of the water management strategy implemented, such as installing dual drainage systems and installing rainwater harvesting systems within the premises. Among other strategies introduced by SPAN for water resource management apart from installing water-efficient pipes, showers, and showerheads, and registering efficient water equipment as has been completed in developed countries such as Singapore. This strategy can contribute to the change in water use habits [18,19].

Socio-demographic factors are high household income associated with energy consumption in homes because of the household's ability to invest in products and improvements to enhance the energy efficiency, such as purchasing new equipment and more energy-efficient equipment technologies [14]. The ability to invest in products means the purchasing power of water-saving devices such as automatic dishwashing machines, washing machines, automatic water pipes, and dual pumping systems. However, Beal et al. [15] mentioned that homes with high-technology water-saving tools do not necessarily have a cost-effective attitude. Saving efforts will not work if reduction action is not reflected in their daily behavior, even if high income, large households, a young age, and more educated people are installing efficient equipment. Therefore, Attari's [20] study suggests that the most effective measures to reduce water use are through reduction behaviors rather than using these technological tools, because of the high cost of having these high-tech water-saving devices such as the use of showers often represent most water use in residential [21,22].

With population growth, economic development, rapid urbanization, and climate change, countries worldwide face water scarcity. Malaysia is blessed with abundant rainfall that contributed to abundant water resources, but inefficient management and water usage abuse have resulted in a water crisis that has caused hardships. For sustainable water resources management, countries worldwide are shifting from supply-based water management to demand-based water management. The policies of the Malaysian government also aim to achieve sustainability of water resources. Malaysia's water consumption rate is 226 liters per day, which is above the recommendation by the United Nations water consumption rate for Individuals is 165 liters per day [23]. The study results by Wai Leng et al. stated that the majority of domestic water consumers in Malaysia do not adopt the practice of saving water [24]. This behavior will lead to water scarcity problems in the future if these bad habits continue. The households should be more sustainable by avoiding wastage of valuable water resources.

Bari et al. found that water consumption among the Klang Valley people is still above average in Malaysia with water consumption per capita of 288 liters per day [25]. Rinching Town, located in the Hulu Langat district of the Klang Valley, was affected by the lack of water supply due to declining water levels in rivers and dams since January 2014, causing some areas in the Klang Valley to be affected by water rationing activities. Therefore, the Rinching Town area residents should save water consumption in their homes to minimize water resources supplied. However, whether the community around the Klang Valley, especially in Rinching Town, is aware of the forms of wastage of water they are doing daily. Therefore, the government's recommendation on consumers is to be thrift in using domestic water supply. However, people's awareness of the waste of water supply for external domestic activities such as washing vehicles, watering trees, and washing corridors is very low. For efficient and sustainable water resources management, it is essential to understand the pattern of water consumption. Usually, water consumption pattern depends on certain socio-economic and climatic factors. Various environmental problems, including water shortages, are partly rooted in human behavior, and can thus be managed by changing behavior to reduce the environmental impacts [26].

Hence, this study was conducted to achieve the following objectives: (1) to analyze the behavior and daily activities of the residents of Rinching Town for domestic water use; (2) to determine the relationship of population demographic factors to household water consumption behaviors; (3) to determine the perception of water-saving based on daily activities and installing water-saving devices in the home. This study's information enables the development of a demographic, psycho-social, and end-use profile that may overestimate or underestimate their use of water. Combining these multiple data sources allows the study to make significant empirical contributions to this field's literature. Besides, the findings could help inform demand management policies such as targeted community education.

## 2. Materials and Methods

This study's data were obtained through surveys generated among households in Rinching Town, which consist of Section 1 until Section 2 of Rinching Town (Figure 1). Rinching Town is 18.47 km$^2$ with 200 household units [27,28]. The study area was selected based on the factor of the population of Rinching Town area was involved in water supply disruption that had to be implemented on the districts in Selangor. The main domestic water supply sources for Rinching Town are the Semenyih River, Lalang River, and Rinching River after the Semenyih Dam treatment [29]. Based on observations, most residential areas in Rinching Town are comprised of two-story and single-story terrace housing areas. Their nature complicates behavioral models that explain voluntary behavior change. As human behavior is typically complex, several factors are not easily controlled, leading to many consequences. The primary research data collected through the survey questions were distributed. The instruments used for this survey were online and on-site surveys. The survey form was distributed through a Google Form, which was a convenient method for

the respondents and data collections from each housing section representative. Focusing on the internal and external influences that need to be taken into account to understand and influencing water conservation behavior [30]. Approaches that model behavior as a function of attitudes, values, habits, and personal norms influence individuals.

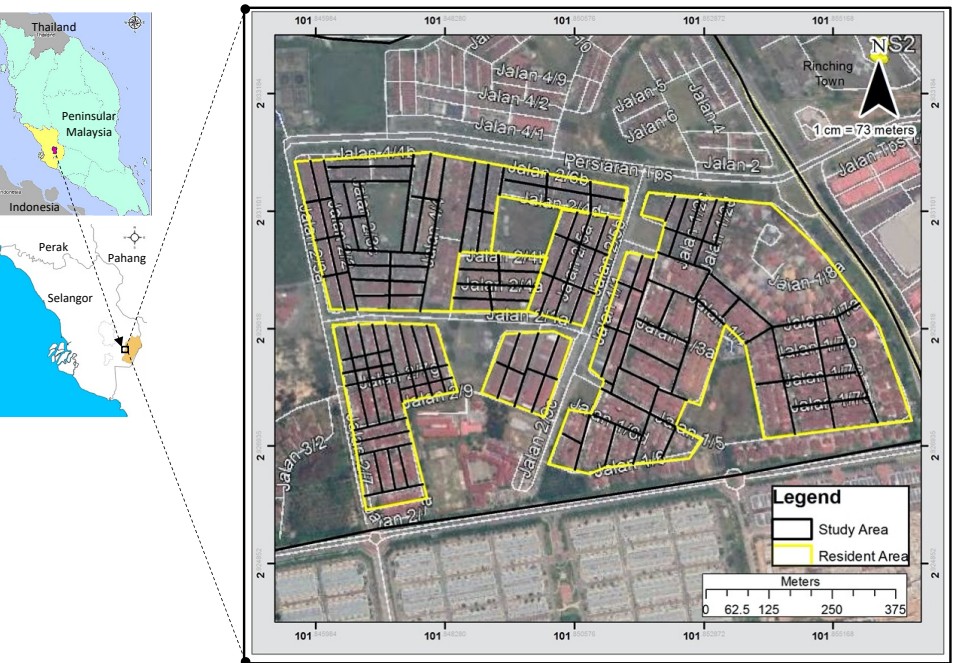

**Figure 1.** Rinching Town location.

Respondents had to fill in information such as age, gender, status, education level, and others. The information from this section was used to divide the respondents according to their categories to ensure that the data obtained were more specific, and the analyzing process was manageable. This section contains 16 objectives questions to see households' daily activity using the highest domestic water sources. The data obtained were also used to determine water use duration in daily activities such as self-cleaning, washing, and watering. Analysis using the Statistical Package for Social Science (SPSS) helps identify mean, frequency, and other analyses such as relationship descriptive analysis, cross-tabulation analysis, and multivariate variation analysis (MANOVA). This study's findings will show whether respondents use domestic water supplies that are supplied to them prudently or not. The IDs in the survey form were pre-processed to create the behavioral analysis to improve the water efficiency which can generally be grouped into three major groups: technical measures, behavioral measures, and, eventually, economic measures. The description for each ID is shown in Table 1.

Several questions on the respondent's behavior involved choices as statements while responding to the question regarding the behavioral analysis and were placed on a 5-point Likert scale with a range of 'never' to 'always'. The 5-point Likert scale with its score method is 'never' with a 1-point score, 'rarely' with a 2-point score, 'sometimes' with a 3-point score, 'often' with a 4-point score, and 'always' with a 5-point score used in this study. The important outputs in interpret analysis based on MANOVA are $p$, df, F, sig., and partial $\eta^2$. $p$ is a statistical significance for the dependent variable, with $p < 0.05$. df stands for degrees of freedom to calculate the statistical significance and represent the size of the samples used. Next, F can be calculated by dividing the mean squares for the variable by its error mean squares and significance level, sig., which gives the probability could have occurred by chance sig. values should be smaller than 0.05 ($p < 0.05$). The partial $\eta^2$ is the magnitude of the effect in samples or as a measure of effect size [31].

**Table 1.** The ID and description of the survey questions.

| ID | Description |
| --- | --- |
| **Technical Measures** | |
| SWLS1 | Shut off tap water, not in use |
| SWLS2 | Monitor and repair leaking water pipes |
| SWLS3 | Using collected rainwater to water plants |
| SWLS4 | Using collected rainwater to wash vehicles |
| SWLS5 | Using water bucket not rubber hose to wash the vehicles |
| SWLS6 | Using the washing machine only when full loads |
| **Behavioral Measures** | |
| SWLS7 | Shut off the tap water while brushing teeth |
| SWLS8 | Watering plants in the early morning and late evening |
| SWLS9 | Hand wash for laundry |
| SWLS10 | Check water pressure is too low or too high. |
| SWLS11 | Check water meter reading used from pipes every month |
| SWLS12 | Check for any open water pipe before leaving the house. |
| **Economic Measures** | |
| SWLS13 | I feel confident in my ability and my family to save water |
| SWLS14 | I want to install more water-saving equipment at home |
| SWLS15 | I do not think it is necessary to install water-saving equipment at home |
| SWLS16 | I feel disturbed if my family member or anyone takes a bath too long |
| SWLS17 | I think it is a requirement to educate the family members or anyone else to save water usage |
| SWLS18 | Rainwater harvesting system is important |

## 3. Results

The sample size was determined by using the sampling methods of Cohen et al. [32,33]. Once the interpolation method was done, the total number of samples required in this study amounted to 132 samples (covering a sampling error of 5%) of the total households in Rinching Town, which has 200 units of households. A total of 135 questionnaires were distributed randomly in the study area. Respondents were assisted to understand each of the actual meanings of the questionnaire. However, after completing the questionnaire's operation in the field, the questionnaire data's pre-analysis process shows that 120 sets of questionnaires have been considered "cleanest", complete, meaningful, and free from technical and human errors.

The instrument's reliability for this study was measured to assess the questionnaire instruments' consistency and accuracy. Reliability analysis is carried out through the Cronbach alpha, which provides an internal measure of consistency for scales of questionnaires and is expressed as numbers between 0 and 1. If the test items are associated with each other, the alpha's value will increase and show a high level of internal consistency [34]. The reliability analysis was conducted with 34 questions and 18 questions based on the Likert Scale, showing that $\alpha$ reliability (Alpha Cronbach) was 0.728 with 34 items. This $\alpha$ explains that the instruments' reliability is high and satisfactory and suitable to achieve the study's objectives.

### 3.1. Demographic Analysis

The first section of surveys is the demography variable of respondents. The frequency and distribution of respondents based on demographic factors are shown in Table 2. According to Table 2, the respondents' highest proportion based on this questionnaire was among respondents aged 20 to 40, who made up almost one-half of the total respondents with 47.5%. Meanwhile, the respondent's lowest frequency was among respondents aged over 60 with nine respondents and comprised 7.5% of total respondents. The proportion of respondents among men was higher than that of females with 66.7% respectively compared to 33.3%, for female respondents. Next, Malays respondents are the highest compared to

other races that comprise 92.5% (111 respondents), in conjunction with the fact that ethnic Malay is the majority population in Rinching Town.

**Table 2.** Respondent distribution based on demographic variables.

| Demography Variables | Respondents | |
|---|---|---|
| | Frequency | Percentage (%) |
| **Age** | | |
| <20 years old | 12 | 10.0 |
| 20 to 40 | 57 | 47.5 |
| 41 to 50 | 21 | 17.5 |
| 51 to 60 | 21 | 17.5 |
| >60 years old | 9 | 7.5 |
| **Gender** | | |
| Male | 80 | 66.7 |
| Female | 40 | 33.3 |
| **Race** | | |
| Malay | 111 | 92.5 |
| Chinese | 2 | 1.7 |
| Indian | 3 | 2.5 |
| Others | 4 | 3.3 |
| **Marital Status** | | |
| Single | 40 | 33.3 |
| Married | 78 | 65.0 |
| Others | 2 | 1.7 |
| **Education Level** | | |
| No formal education | 1 | 0.8 |
| Primary School | 4 | 3.3 |
| Secondary School | 42 | 35 |
| STPM/Matriculation | 15 | 12.5 |
| University | 58 | 48.3 |
| **Number of Households** | | |
| 1 to 3 members | 26 | 21.7 |
| 4 to 6 members | 75 | 62.5 |
| 7 to 9 members | 18 | 15.0 |
| 10 to 12 members | 1 | 0.8 |
| **Household Income** | | |
| <RM 2500 | 20 | 16.7 |
| RM 2500 to RM 4000 | 46 | 38.3 |
| RM 4001 to RM 6000 | 21 | 17.5 |
| RM 6001 to RM 8000 | 15 | 12.5 |
| RM 8001 to RM 10,000 | 10 | 8.3 |
| >RM 10,000 | 8 | 6.7 |

Besides, married respondents accounted for 65% of the total respondents, followed by single respondents of 33.3%. On the other hand, most of the respondents were university graduates of 48.3%, followed by high school graduates of 35%. The respondents with the highest number of household members are 4 to 6 people in a residential home in Rinching Town and constitute 62.5% of the sample. Meanwhile, the lowest number of households was 1 to 3 people accounting for 21.7% of the sample and 7 to 9 people accounting for 15%. Finally, respondents with households earning RM 2500 to RM 4000 were the highest (38.3%) in this survey followed by respondents with households of RM 4001–RM 6000 (17.5%) and distinguished 0.8% of households earning less than RM 2500 (16.7%).

### 3.2. Behavioral Analysis of Domestic Water Usage of Households

Among the best water demand management strategies, such as domestic water, is to understand the water users' needs and habits. Implementation of demand management and water conservation strategies that is effective and relevant is strongly underpinned by an understanding and knowledge of how consumers perceive and use water [35,36]. The respondents were asked to choose the appropriate answers. The data obtained were analyzed using descriptive statistics. Thus, the descriptive analysis helped investigate the mean value, standard deviation, and variance of the data set obtained as depicted in Table 3, and the mean score (%) was shown in Figure 2.

**Table 3.** Behavioral usage of domestic water of household.

| ID | Mean | | Standard Deviation | Variance |
|----|------|------|----|----|
| | Statistic | Std. Error | | |
| SWLS1 | 4.45 | 0.095 | 1.043 | 1.090 |
| SWLS2 | 3.42 | 0.111 | 1.220 | 1.490 |
| SWLS3 | 2.20 | 0.125 | 1.369 | 1.876 |
| SWLS4 | 2.07 | 0.125 | 1.379 | 1.902 |
| SWLS5 | 2.45 | 0.114 | 1.255 | 1.577 |
| SWLS6 | 3.53 | 0.120 | 1.321 | 1.747 |
| SWLS7 | 3.87 | 0.112 | 1.236 | 1.528 |
| SWLS8 | 2.86 | 0.125 | 1.373 | 1.887 |
| SWLS9 | 2.31 | 0.101 | 1.115 | 1.243 |
| SWLS10 | 2.87 | 0.112 | 1.229 | 1.511 |
| SWLS11 | 2.53 | 0.111 | 1.222 | 1.495 |
| SWLS12 | 3.97 | 0.105 | 1.151 | 1.327 |

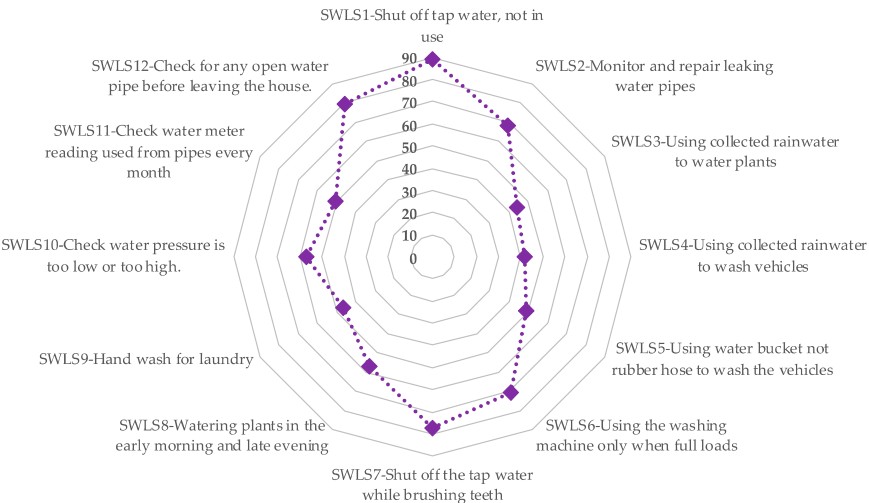

**Figure 2.** Technical and behavioral response to saving water.

Understanding household behavior, activities, and how this relates to water needs will make it possible to reduce wasteful behavior by increasing the knowledge and adaptive capacity of water users [37]. The analysis showed that 89% of Rinching Town residents have a high awareness to save domestic water for most fundamental saving behaviors such as shutting off the water pipes when not using (SWLS1 with mean = 4.45, standard deviation = 1.043, and variance = 1.090) and also while brushing their teeth (SWLS7 with mean = 3.87, standard deviation = 1.236, and variance = 1.528). However, their water-saving awareness is average for other types of domestic water use behaviors with using collected rainwater to wash the vehicles are the most uncommon practices for the household (SWLS4 with mean = 2.07, standard deviation = 1.379, and variance = 1.902). However, with the mean values of 2.20 and 2.07 for rainwater use for tree watering (SWLS3) and car washing

(SWLS4), respectively, and less than 50% of households in Rinching Town use rainwater as an alternative to savings. Furthermore, monitoring behaviors to prevent domestic water wastage can be found in three statements: monitoring and repairing leaky water pipes, monitoring and repair pipelines in case of water pressure from low or too high pipes, and monitoring the water meter readings used from each pipe. The mean analysis of these three statements showed that respondents were on a 'sometimes' scale for SWLS2 (mean = 3.42) and on a 'rarely' scale for SWLS10 (mean = 2.87) and SWLS 11 (mean = 2.53) in performing monitoring actions.

Behaviors that reflect the savings and 'often' of respondents saving water is an act of checking whether there is an open water pipe before leaving the house (SWLS12) with mean value = 3.97, standard deviation = 1.151, and variance value = 1.327. However, other behaviors such as car wash using a water bucket instead of rubber hose (SWLS5), watering in the morning and late afternoon (SWLS8) and hand washing (SWLS9) indicate that respondents rarely do this. In essence, the people of Rinching Town have shown a moderate tendency to adopt economic behaviors in the use of domestic water in their homes based on descriptive analysis of mean values that tend to be 'sometimes' scaling in savings except for a basic act of saving water that closes the water pipe when not in use and when brushing teeth shows a large 'often' scale.

The finding that most respondents perform almost all activities could indicate that these behaviors are habitual or routine. It could also mean that behaviors, such as washing a full laundry load, are economically beneficial since large appliances normally use high energy and water volume in the house. An alternative source that easily available is the rainwater harvesting method. Malaysia, which is located in the tropical climate, has received very high rainfall every year which is about 2400 mm, so the country's residents should use rainwater as one of the alternative water sources in overcoming the problem of water shortages. Although this rainwater is not used as drinking water, many other activities can be completed, such as watering trees, washing cars, floor washing, and toilet showers. This behavior can save on the domestic clean water supply.

The duration and amount of domestic water use in the home's daily activities play a role in determining whether domestic water-saving behaviors exist [38]. Randolph and Troy [39] proposed 13 actions for reducing household water consumption such as taking shorter showers, filling the washing machine before using, reducing garden watering, and reducing car washing; it was shown that almost all of these actions are to some extent efficient. Figure 3 shows the frequency and percentage of water consumption in daily activities by households. Figure 3 shows that most of the respondents washed the dishes more than three times, with 51% choosing the same with the frequency of 61 times. Whereas for toilet use, the percentage value for use was more than three with 67%, and a frequency of 80. Most households in Rinching Town also take baths, with 53% of them taking 6 to 10 min to shower. For washing clothes using washing machines, 57% of respondents would only use the washing machine once a day compared to using the washing machine three times or more with 6% and 7%, respectively. Washing clothes with fully loaded washing machines are more efficient than manual clothes-washing. Even among the families using a washing machine, water consumption is relatively high. Jiang's findings indicated that a large percentage of household water was used for flushing toilets, personal hygiene uses, and washing clothes [40]. Households in Rinching Town also only clean their vehicles at home more than once a week (38%), and the percentage is also high, with 26% on an average weekly car wash with a frequency of 31. For domestic water use activities for plant watering, most households water once a day with a percentage of 59% with a frequency of respondents being 71.

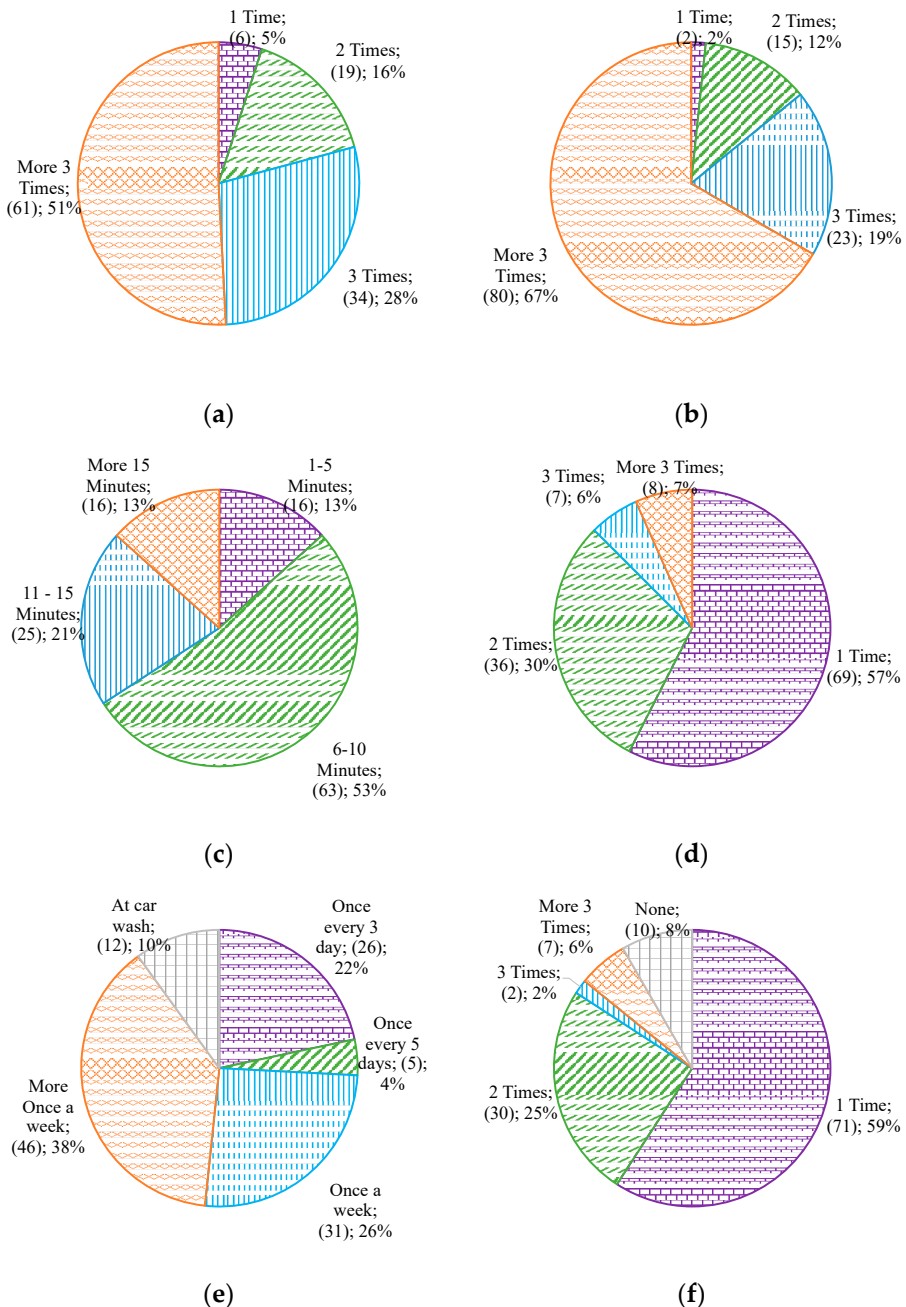

**Figure 3.** Water consumption activities, the values (description, frequency, percentage): (**a**) Dishwashing; (**b**) Toileting; (**c**) One-time showering; (**d**) Washing clothes; (**e**) Washing car; (**f**) Watering plants.

### 3.3. Relationship between Demographic Factors and Behavioral Usage of Domestic Water

Water consumption patterns and behaviors are highly varied amongst households due to the influencing factors of climate, socio-demographics, house size, family composition, water appliances, cultural and personal practices [19]. The bivariate correlation analysis was used to determine the relationship between demographical variables and behavioral usage of water. Analysis of the MANOVA was performed to look at the relationships between these demographic characteristics in more detail and based on Pearson's bivariate correlation analysis analyzed in Appendix A. The basic analysis of Pearson's bivariate correlation in Appendix A shows that the behavior of using rainwater for washing machines (SWLS4) was found to be related to age demographic characteristics. Therefore, further analysis of the MANOVA was performed to look at these two relationships in more detail. The $F$ ($df1$, $df2$) is the list of the degrees of freedom used in determining the F statistics.



Table 4 shows a test of the effects of subjects showing no significant difference between the subgroups of age variables on domestic water use rainwater for washing machines (SWLS4) with $F (4, 115) = 1.253$, $p = 0.293$ and partial values $\eta^2 = 0.042$.

**Table 4.** Tests of between-subjects' effect (age).

| Source | ID | Sum of Squares | df | Mean Square | F | Sig. (p) | Partial Eta Squared ($\eta^2$) |
|--------|------|----------------|-----|-------------|-------|----------|-------------------------------|
| Age | SWLS4 | 9.449 | 4 | 2.362 | 1.253 | 0.293 | 0.042 |
| Error | SWLS4 | 216.876 | 115 | 1.886 | | | |

The basic analysis of Pearson's bivariate correlation, as shown in Appendix A, demonstrates a significant relationship between washing machine use when full load (SWLS6) and handwashing behavior (SLWS9) with gender. Table 5 shows a test of the effect of subjects showing significant differences between genders on water consumption behavior as seen in the behavior of washing machines with a full load (SWLS6) with $F (1, 118) = 5.467$, $p = 0.021$, partial value $\eta^2 = 0.044$ as well as hand washing (SWLS9) behavior with $F (1, 118) = 6.481$, $p = 0.012$, partial $\eta^2 = 0.052$.

**Table 5.** Tests of between-subjects' effect (gender).

| Source | ID | Sum of Squares | df | Mean Square | F | Sig. (p) | Partial Eta Squared ($\eta^2$) |
|--------|-------|----------------|-----|-------------|-------|----------|-------------------------------|
| Gender | SWLS6 | 9.204 | 1 | 9.204 | 5.467 | 0.021 | 0.044 |
| | SWLS9 | 7.704 | 1 | 7.704 | 6.481 | 0.012 | 0.052 |
| Error | SWLS6 | 198.663 | 118 | 1.684 | | | |
| | SWLS9 | 140.262 | 118 | 1.189 | | | |

The race's demographic characteristics also had a significant relationship with washing machines when fully loaded (SWLS6) based on a preliminary analysis of bivariate correlation. Table 6 shows a test of the effect of subjects showing significant differences in races on water consumption behavior that can be seen in the behavior of washing machines when a full load (SWLS6) with a value of $F (3, 116) = 3.490$, $p = 0.018$ and partial $\eta^2 = 0.083$.

**Table 6.** Tests of between-subjects' effect (race).

| Source | ID | Sum of Squares | df | Mean Square | F | Sig. (p) | Partial Eta Squared ($\eta^2$) |
|--------|-------|----------------|-----|-------------|-------|----------|-------------------------------|
| Race | SWLS6 | 17.207 | 3 | 5.736 | 3.490 | 0.018 | 0.083 |
| Error | SWLS6 | 190.660 | 116 | 1.644 | | | |

The marital status also showed correlations between three types of domestic water consumption behaviors, such as using rainwater to water the tree (SWLS3), washing the vehicle (SWLS4), and using a washing machine when fully loaded (SWLS9). Table 7 shows a test of the effect of subjects showing no statistically significant differences in marital status on water use behavior seen in behavior using rainwater to water the principal (SWLS3) with $F (2, 117) = 2.451$, $p = 0.091$ value and partial $\eta^2 = 0.040$. Meanwhile, the behavior of using rainwater to wash the vehicle (SWLS3) with $F$ value $(2, 117) = 3.391$, $p = 0.037$, partial $\eta^2 = 0.055$, and handwashing (SWLS9) with $F (2, 117) = 2.369$, the value of $p = 0.098$ and the partial $\eta^2 = 0.039$.

**Table 7.** Tests of between-subjects' effect (marital status).

| Source | ID | Sum of Squares | df | Mean Square | F | Sig. (p) | Partial Eta Squared ($\eta^2$) |
|---|---|---|---|---|---|---|---|
| Marital Status | SWLS3 | 8.976 | 2 | 4.488 | 2.451 | 0.091 | 0.040 |
| | SWLS4 | 12.401 | 2 | 6.201 | 3.391 | 0.037 | 0.055 |
| | SWLS9 | 5.758 | 2 | 2.879 | 2.369 | 0.098 | 0.039 |
| Error | SWLS3 | 214.224 | 117 | 1.831 | | | |
| | SWLS4 | 213.924 | 117 | 1.828 | | | |
| | SWLS9 | 142.208 | 117 | 1.215 | | | |

The educational level also shows a correlation with domestic water consumption behavior, namely checking for open water pipes before leaving the house (SWLS12). Table 8 shows a test of the effect of subjects showing significant differences in education level on water consumption behavior that can be seen in the behavior of checking for open water pipes before leaving the house (SWLS12) with an *F* value of (4, 115) = 2.486, *p* = 0.047 and partial $\eta^2$ = 0.080.

**Table 8.** Tests of between-subjects' effect (education level).

| Source | ID | Sum of Squares | df | Mean Square | F | Sig. (p) | Partial Eta Squared ($\eta^2$) |
|---|---|---|---|---|---|---|---|
| Education Level | SWLS12 | 12.565 | 4 | 3.141 | 2.486 | 0.047 | 0.080 |
| Error | SWLS12 | 145.302 | 115 | 1.263 | | | |

The analyses' demographic factors are age, gender, race, marital status, education level, number of households, and their respective monthly income. The Pearson bivariate correlation analysis was conducted at the first level to filter out the unrelated demographic factor that found out demographic factors, which are the number of households and income, has no respective relationship with any of 12 different behavioral usages of domestic water. Further analysis using multivariate analysis (MANOVA) was conducted and summarized in Table 9.

Young people tend to perform saving measures of using rainwater to wash their vehicles (SWLS4) compared to elder ones based on significant differences in mean values of them having the highest mean value (mean = 2.50) followed by respondents with an age of 20 to 40 years old (mean = 2.23), 41 years to 50 years (mean = 2.05), 51 years to 60 years (mean = 1.67), and more than 60 years old respondents (mean = 1.89). However, the pattern of age effect varied across behaviors and studies. In some studies, the effect was not even significant [41]. Women are seen to have a higher tendency to perform saving measures (mean = 3.93 and 2.68) using a washing machine when a full load (SWLS6) and hand washing for laundry (SWLS9), respectively, compared to men (mean = 3.34 and 2.14). A usual finding is that women are more pro-environment than men, similar to Dietz et al., which found that women were more likely to consume in a pro-environmental manner than men [41].

Chinese respondents tended to implement saving behavior using a washing machine only when full (SWLS6) than other races. On the other hand, single people had the highest mean (mean = 2.58, 2.53 and 2.63) compared to those who were married (mean = 2.03, 1.86 and 2.17) and others then showed that they had a high tendency towards saving actions on these three behaviors which are water consumption behavior by using rainwater to water the trees (SWLS3), washing the vehicle (SWLS4) and hand laundry (SWLS9). The level of education with the habits of checking if there is an open water pipe before leaving the house (SWLS12) showed that university graduates have the highest tendency (mean = 4.19), followed by high school leavers (mean = 3.95), STPM/Matriculation/Teaching College (mean = 3.53), and primary school (mean = 2.75).

**Table 9.** Mean difference estimation.

| ID | Variables | Mean | Std. Error | 90% Confidence Interval | |
|---|---|---|---|---|---|
| | | | | Lower Bound | Upper Bound |
| | | | Age | | |
| SWLS4 | <20 years old | 2.50 | 0.396 | 1.715 | 3.285 |
| | 20–40 | 2.23 | 0.182 | 1.868 | 2.588 |
| | 41–50 | 2.05 | 0.300 | 1.454 | 2.641 |
| | 51–60 | 1.67 | 0.300 | 1.073 | 2.260 |
| | >60 years old | 1.89 | 0.460 | 0.977 | 2.801 |
| | | | Gender | | |
| SWLS6 | Male | 3.34 | 0.145 | 3.050 | 3.625 |
| | Female | 3.93 | 0.205 | 3.519 | 4.331 |
| SWLS9 | Male | 2.14 | 0.122 | 1.896 | 2.379 |
| | Female | 2.68 | 0.172 | 2.334 | 3.016 |
| | | | Race | | |
| SWLS6 | Malay | 3.57 | 0.122 | 3.327 | 3.809 |
| | Chinese | 5.00 | 0.907 | 3.204 | 6.796 |
| | Indian | 3.67 | 0.740 | 2.201 | 5.133 |
| | Others | 1.75 | 0.641 | 0.480 | 3.020 |
| | | | Marital Status | | |
| | Single | 2.58 | 0.214 | 2.151 | 2.999 |
| SWLS3 | Married | 2.03 | 0.153 | 1.722 | 2.329 |
| | Others | 1.50 | 0.957 | −0.395 | 3.395 |
| SWLS4 | Single | 2.53 | 0.214 | 2.102 | 2.948 |
| | Married | 1.86 | 0.153 | 1.556 | 2.162 |
| | Others | 1.50 | 0.956 | −0.394 | 3.394 |
| SWLS9 | Single | 2.63 | 0.174 | 2.280 | 2.970 |
| | Married | 2.17 | 0.125 | 1.919 | 2.414 |
| | Others | 2.00 | 0.780 | 0.456 | 3.544 |
| | | | Education Level | | |
| SWLS12 | No formal education | 3.00 | 1.124 | 0.773 | 5.227 |
| | Primary School | 2.75 | 0.562 | 1.637 | 3.863 |
| | Secondary School | 3.95 | 0.173 | 3.609 | 4.296 |
| | STPM/Matriculation | 3.53 | 0.290 | 2.958 | 4.108 |
| | University | 4.19 | 0.148 | 3.897 | 4.482 |

Thus, demographic characteristics are identified to affect domestic water consumption behavior and the average water bill [4,5]. Young people, bachelors, women, and university graduates are seen to have the kind of saving measures such as using washing machines only when full loads, using rainwater to water the trees and washing the vehicle, washing clothes by hand checking open pipes before leaving home based on the mean significant difference. This analysis further demonstrates that women and groups with high education levels tend to practice water-saving attitudes [6].

### 3.3.1. Relationship of Demographic Features and Average Monthly Water Bills

Bivariate correlation analysis was also used to determine the importance of demographic characteristics on average monthly water bills by households in Rinching Town. The bivariate correlation procedure calculates the unification relation for a set of variables and displays the matrix form results. This analysis is useful in determining the strength and direction of associations between two scales or ordinal variables.

Table 10 shows that the demographic factors affecting the number of households influence the average monthly water bill by households in Rinching Town compared to other factors such as age, gender, race, education level, and household income estimate with a correlation value of 0.403. This value of 0.403 suggests a significant and positive correlation between households and the average monthly water bill.

**Table 10.** Bivariate correlation.

| Monthly Water Bill | Age | Gender | Race | Education Level | Number of Households | Household Income |
|---|---|---|---|---|---|---|
| Pearson Correlation | −0.028 | 0.011 | −0.117 | −0.121 | 0.403 ** | 0.030 |
| Sig. (2-tailed) | 0.762 | 0.908 | 0.203 | 0.186 | 0.000 | 0.747 |
| *N* | 120 | 120 | 120 | 120 | 120 | 120 |

** Correlation is significant at the 0.01 level (2-tailed).

Demographic factors such as age, race, and education level had negative correlation values with −0.028, −0.117, and −0.121 indicating that these factors did not influence household average water bill. Education is also a demographic factor that does not impact the average monthly water bill in Rinching Town. Demographic factors such as gender and household income estimates may have a positive correlation value, but they are also less affected because they are far from 1. This study further found that as income level increases, the participation of water conservation higher. Several studies have found a relationship between higher income levels and support for environmental causes and environmental behaviors [42–44]. However, it should be noted that this does not imply that higher income individuals are more environmentally concerned, just that they may have the time or resources available to commit to performing environmental behaviors.

The finding that income correlates with behavior is interesting when we consider prior research. Specifically, Yu Xue et al. [42] and Juana et al. [44] found no relationship between income and environmental behavior or a negative relationship. The findings in this paper suggest that as income increases, water conservation behaviors increase, which is similar to the findings by Peng Xue et al. [45].

### 3.3.2. Relationship between the Number of Households and Monthly Water Bills

The population size and the number of households affects the amount of water consumption [46]. If there is an increase in household and population size, water consumption will also increase [47]. The number of households is vital in determining the average monthly water bills per month and energy management. The demographic features focused on this section are the number of households as this feature has a significant relationship to the monthly water bill. The cross-tabulation analysis used to prove the relationships is depicted in Figure 4.

Households with 4 to 6 family members were the highest respondents in the survey, with a percentage of 62.5%. It was followed by 1 to 3 members (21.7%), 7 to 9 households (15.0%), and 10 to 12 members (0.8%). Table 2 suggests briefly that the higher the number of households, the higher the monthly water bills per month for a dwelling. For example, the low number of households (1–3 members) had the highest percentage (65.4%) for the average monthly water bills of less than RM 20.00 and had a low percentage (7.7%) for an average water bill above RM 41.00. For a moderate number of households (4 to 6 people) 31 out of 75 respondents have an average water bill between RM 21.00 to RM 30.00. The average water bill is acceptable if compared to the average number of family members. 22.7% of households with 4 to 6 members prove that they use little water an average water bill of less than RM 20.00.

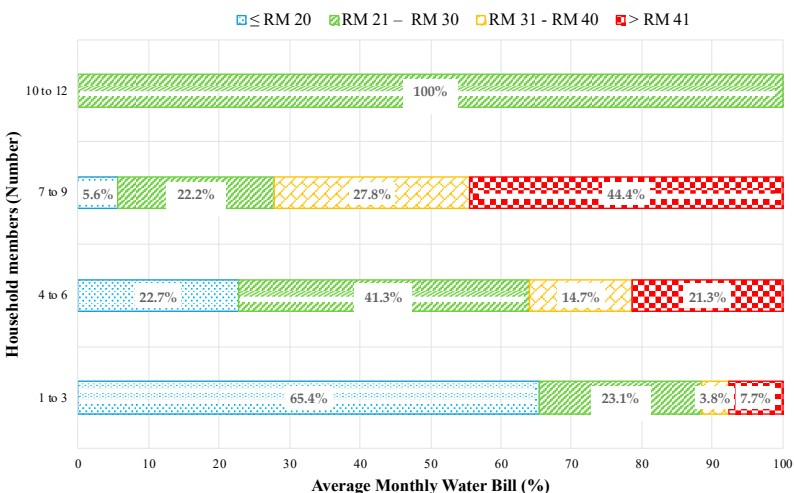

**Figure 4.** Cross-tabulation analysis of average monthly water bill.

In contrast to the high number of households with a large number of people (7 to 9 people), 44.4% of households will have an average monthly water bill of more than RM 41.00, and only 5.6% will have an average water bill per month less than RM 20.00. Hence, the number of households is associated with the average water bill within. Thus, the higher the number of family members, the higher the average water bill in a month.

This result shows that household differences are essential as the amount of water consumption is different. In this study, the relationship between determinants has become limited, as the participating households are mostly elderly. Thus, the result of a study based on population selection among senior citizens alone may not represent the community as a whole. However, in contrast to the study of Rathnayaka et al. [48] found that age factor is not a decisive factor given that the presence of children between the ages of 12 and 18 years is an indispensable factor in explaining the use of water by households.

Simultaneously, socio-demographic characteristics such as the amount of income, sex, and education level are also essential to be a highlight in the effort of saving water [40]. High-income households tend to use more water tap than low-income households, and households' usage of a large amount of water for baths comprise high-growth households with teenagers and children [6]. Socio-economic factors also play an essential role when studies have found that women, low-income households, and high education levels tend to practice water-saving attitudes.

### 3.4. Perception of Water-Saving Based on Daily Activities and Water-Saving Devices

The usage of saving and water efficiency technologies such as low-pressure water pipe water, showerheads, high-powered washing machines, and rainwater tanks are seen as part of water resource management methods towards the sustainability approach. The respondents were asked which factors affect their ability to implement water-saving technologies in their dwellings. To highlight these issues, Table 11 and Figure 5 show the perception of households on water-saving based on the daily activities and water-saving devices for this analysis.

In total, 83.6% of respondents agree that they are confident with their ability and family to save water both indoors and outdoors (mean = 4.18 and standard deviation = 0.984) and feel that it is a requirement to educate family members or anyone else to save the water (mean = 4.23, standard deviation = 0.991). For SWLS16, 69.4% of respondents are uncertain (mean = 3.47, standard deviation = 1.144) whether to assume that it is a disruption if their family members or anyone baths take too much time for bathing. For the uses of water-saving equipment, 50.6% of respondents are also uncertain (mean = 2.53, standard deviation = 1.068) to install the equipment in their homes, but 84.6% agree to consider this water-saving method as a necessity to save water use at home. This result is further evident when 79.8% agree that the rain harvesting system is essential to save

domestic water use (mean = 3.99, standard deviation = 0.999). Therefore, it was known that residents in Rinching Town have good awareness and were confident in saving domestic water use. They also agree that they can save water if they run the proper alternatives, such as using water-saving equipment and rainwater harvesting systems.

**Table 11.** Perception of water-saving based on daily activities and water-saving devices.

| ID | Mean | | Standard Deviation | Variance |
|---|---|---|---|---|
| | **Statistic** | **Standard Error** | | |
| SWLS13 | 4.18 | 0.089 | 0.984 | 0.969 |
| SWLS14 | 3.48 | 0.098 | 1.076 | 1.159 |
| SWLS15 | 2.53 | 0.097 | 1.068 | 1.142 |
| SWLS16 | 3.47 | 0.104 | 1.144 | 1.310 |
| SWLS17 | 4.23 | 0.091 | 0.999 | 0.999 |
| SWLS18 | 3.99 | 0.091 | 0.999 | 1.000 |

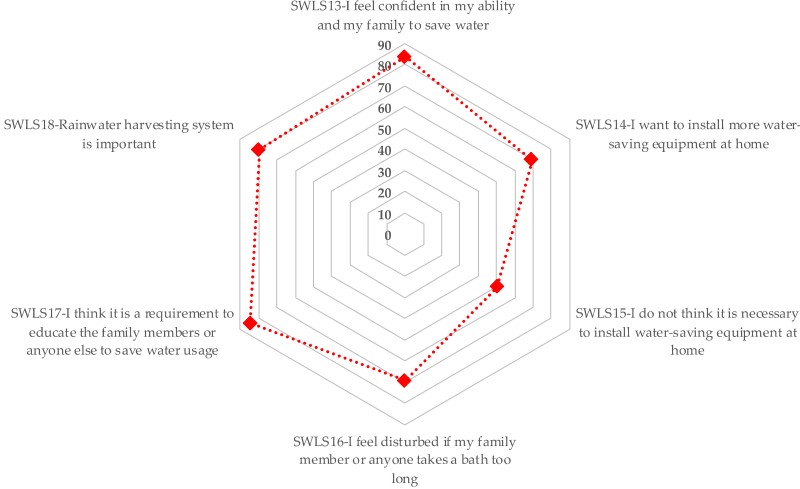

**Figure 5.** Ability to implement water-saving technologies and devices.

*3.5. Bivariate Correlation Analysis of Water-Saving Perceptions*

This analysis reported stronger habitual water-saving behaviors and reported the requirement to educate the household to save water. This finding is particularly important because it demonstrates the importance of water conservation habits. Based on Table 12, the strong correlation between the ability to conserve water indoors and outdoors with the need to educate family members or anyone to conserve water use (SWLS 17) is 0.523, and to realize the importance of using rainwater systems (SWLS18) is 0.505 with Pearson correlation values greater than 0.50.

**Table 12.** Bivariate correlation analysis.

| ID | | SWLS 17 | SWLS18 |
|---|---|---|---|
| SWLS13 | Pearson Correlation | 0.523 ** | 0.505 ** |
| | Sig. (2 tailed) | 0.000 | 0.000 |
| | *N* | 120 | 120 |

** Correlation is significant at the 0.01 level (2-tailed).

## 4. Conclusions

Numerous studies have been carried out so far in various parts of the world to understand how different factors influence water consumption. However, only a few studies have been done on the estimation of water demand in Malaysia. This study's findings have

implications for both policymakers and academicians with the highlighted importance of behavioral determinants of water-saving and water-use behavior. Household water use is the most important indicator for water conservation. Therefore, the habitual water-wasting behavior would be of benefit in affecting household water use. This study has shown that the behavioral, socio-demographic, and contextual variables all have a role in determining household water-saving. Household size and income are important determinants of water use and are out of the control of policymakers. However, this study clearly shows the behavioral factors are significant determinants of water-saving and household water use. This study on the analysis of domestic household water consumption behavior in Rinching Town summarizes the importance of implementing austerity measures in domestic water consumption behavior, how to use water in daily activities, related demographic characteristics with the behavior of domestic water use, as well as households' perception toward water-saving attitudes and equipment. The analysis of behavioral water consumption and daily water usage activities was conducted using descriptive analysis that highlights whether a household in Rinching Town practices water-saving behavior in their daily lives.

More than 50% of respondents practice acceptable water consumption in their daily activities, but they have average water-saving attitudes with different types of water-using behaviors. Multivariate Analysis (MANOVA) suggests that the numbers of households and their respective income have proven to have no relationship to water-saving behavior. While the other hand, the estimation of mean difference showed, demographic factors such as age, gender, race, marital status, and education level are proven to have such a relationship with young people, bachelors, women, and university graduates seen to have some kind of water-saving attitudes. Lastly, more than 50% of the household had strong perceptions of using water-saving devices, and 83.6% of the household had determined to save domestic water consumption, thus proving the community's relationship in the preservation of water resources.

The survey responses provided insight for implementing effective domestic water consumption and saving. The qualitative and quantitative analysis provided in the previous sections should highlight opportunities and conceptual approaches resulting in improved water use habits. Important findings include the following:

(a)  Household showing interest in trying new devices, suggesting a strong interest for further conservation practices;
(b)  Household willingness to change their habits in saving domestic water consumptions. Changes in water use habits had a direct influence on the participants' perception of savings on their water bill;
(c)  The education level of the participants had no significant effect on water savings; and
(d)  Installed water-saving technology contributed to the changes in households' consumption of domestic water.

Initial measures for the saving of domestic water consumption are starting from individual awareness. Although the results showed that household awareness levels were at a high level, it did not fully affect the household behavior level, leading to implementing sustainable practices on their house. This study concluded that a high level of knowledge and a positive attitude towards implementing sustainable principles do not necessarily guarantee a high level of behavior among households. External factors such as enforcement and appreciation are needed in the implementation of sustainable principles. However, it is undeniable that a person with better behavior will act more effectively in a given situation. Changing human habits requires time and resources to build new habits, whereas water scarcity is a current and existing concern in Malaysia. Therefore, both long-term and short-term plans should be considered in this situation.

Several countries have promoted rebate programs for the installation of water-efficient technologies. Currently, Malaysia offers rebates for a series of efficient products, including rainwater tanks, dual flush toilets, and a water-efficient showerhead. Nevertheless, there is still a lack of public response, strict rules and regulations, and suitable government and public policies. Therefore, this study's findings would be useful for water demand

management such as reducing leakages and non-revenue water, raising public awareness on water conservation, and might be useful for introducing new policies to conserve water through efficiency. However, the present study recommends promotion for water-efficient equipment, a behavioral-based approach for conserving water, and develops special training for future generations for potential reduction of water consumption in Malaysia. The only way to solve the problem is to transform our technology, behavior, and way of life.

However, most of these water-saving appliances/devices are rather expensive for the majority of the population. It means that to encourage people to apply water-saving devices and thereby affect the general lowering of water use, subsidizing these technologies is required. The government should provide financial incentives and specialist assistance for the implementation of domestic water-saving technologies. The installation of the technology system involved substantial-high costs. Hence, this may cost the household to install these environmentally friendly products in their homes.

Further studies on a wider area and various types of residences such as terraced houses, bungalow houses, and apartments, can be achieved. The determination of types of residences given that household size affects energy consumption such as domestic water supply. More studies are required to investigate the impact of each water-saving device in improving water efficiency. The number of water-saving equipment and energy efficiency tools can contribute to water consumption habits changes. Therefore, the study based on a comparative analysis of domestic water-saving devices' type and volume against the average water bill per month can be conducted.

**Author Contributions:** Conceptualization, all authors; methodology, H.H.H., and S.F.M.R.; formal analysis, H.H.H., N.H.M.R. and S.F.M.R.; resources, H.H.H.; data curation, H.H.H., N.H.M.R., and S.F.M.R.; writing—original draft preparation, H.H.H., and N.H.M.R.; writing—review and editing, all authors; visualization, H.H.H.; supervision, S.F.M.R.; funding acquisition, S.F.M.R. All authors have read and agreed to the published version of the manuscript.

**Funding:** This research was funded by the Ministry of Education (MOE) Malaysia research grant FRGS/1/2018/TK01/UKM/02/2 and *Dana Padanan Kolaborasi* grant (No: DPK-2020-002) by UKM.

**Institutional Review Board Statement:** Don't need this approval because no involved medical. This study consists of general questionnaires.

**Informed Consent Statement:** Informed consent was; obtained from all subjects involved in the study.

**Data Availability Statement:** All relevant data are in the paper.

**Acknowledgments:** We would like to acknowledge the MOE Malaysia and Universiti Kebangsaan Malaysia for supporting this research to be completed successfully.

**Conflicts of Interest:** The authors declare no conflict of interest.

## Appendix A

Bivariate Pearson Correlation analysis was performed to determine the level and direction of the relationship between ID and independent variables (demographical characteristics). This analysis is conducted to see if there is a relationship of demographic characteristics to domestic water consumption behavior.

**Table A1.** Pearson's bivariate correlation analysis of domestic water consumption and demographic characteristics.

| ID | Test | Age | Gender | Race | Status | Education Level | No. of Household | Income |
|---|---|---|---|---|---|---|---|---|
| SWLS1 | Pearson | 0.137 | −0.034 | −0.051 | 0.002 | 0.069 | −0.118 | 0.034 |
| | Sig. (2-tailed) | 0.136 | 0.712 | 0.577 | 0.986 | 0.455 | 0.198 | 0.709 |
| SWLS2 | Pearson | 0.135 | −0.087 | −0.060 | 0.029 | 0.140 | −0.059 | 0.036 |
| | Sig. (2-tailed) | 0.141 | 0.343 | 0.512 | 0.749 | 0.126 | 0.520 | 0.698 |
| SWLS3 | Pearson | −0.130 | 0.000 | −0.088 | −0.201 * | 0.060 | 0.079 | −0.071 |
| | Sig. (2-tailed) | 0.157 | 1.000 | 0.338 | 0.028 | 0.516 | 0.388 | 0.443 |

**Table A1.** *Cont.*

| ID | Test | Age | Gender | Race | Status | Education Level | No. of Household | Income |
|---|---|---|---|---|---|---|---|---|
| SWLS4 | Pearson | −0.202 * | 0.090 | −0.034 | −0.233 * | 0.022 | 0.129 | −0.038 |
| | Sig. (2-tailed) | 0.027 | 0.328 | 0.712 | 0.011 | 0.815 | 0.159 | 0.681 |
| SWLS5 | Pearson | −0.103 | 0.028 | 0.075 | −0.119 | −0.047 | 0.029 | 0.001 |
| | Sig. (2-tailed) | 0.264 | 0.759 | 0.417 | 0.197 | 0.607 | 0.757 | 0.994 |
| SWLS6 | Pearson | 0.019 | 0.210 * | −0.179 * | −0.022 | −0.085 | 0.173 | 0.094 |
| | Sig. (2-tailed) | 0.833 | 0.021 | 0.050 | 0.812 | 0.357 | 0.059 | 0.305 |
| SWLS7 | Pearson | 0.174 | −0.024 | 0.018 | 0.067 | 0.051 | 0.056 | −0.045 |
| | Sig. (2-tailed) | 0.058 | 0.795 | 0.845 | 0.468 | 0.581 | 0.545 | 0.623 |
| SWLS8 | Pearson | −0.005 | 0.009 | −0.021 | 0.102 | 0.154 | 0.030 | 0.103 |
| | Sig. (2-tailed) | 0.955 | 0.926 | 0.819 | 0.267 | 0.094 | 0.741 | 0.265 |
| SWLS9 | Pearson | −0.106 | 0.228 * | 0.056 | −0.195 * | 0.172 | 0.023 | −0.175 |
| | Sig. (2-tailed) | 0.248 | 0.012 | 0.542 | 0.033 | 0.060 | 0.806 | 0.056 |
| SWLS10 | Pearson | 0.033 | −0.096 | 0.062 | −0.055 | 0.158 | 0.024 | 0.002 |
| | Sig. (2-tailed) | 0.719 | 0.296 | 0.502 | 0.548 | 0.085 | 0.797 | 0.984 |
| SWLS11 | Pearson | −0.047 | −0.034 | 0.080 | −0.120 | 0.009 | 0.056 | 0.035 |
| | Sig. (2-tailed) | 0.611 | 0.713 | 0.382 | 0.193 | 0.923 | 0.540 | 0.702 |
| SWLS12 | Pearson | −0.003 | 0.067 | −0.109 | −0.047 | 0.186 * | −0.002 | −0.070 |
| | Sig. (2-tailed) | 0.977 | 0.469 | 0.237 | 0.606 | 0.041 | 0.980 | 0.445 |

* Correlation is significant at 0.05 level (2-tailed).

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
