# Peer review of "Does the Household Save Water? Evidence from Behavioral Analysis"

_sustainability, doi:10.3390/su13020641_

Round 1

Reviewer 1 Report

Although there are already some studies in this area (some findings are, in some ways expected) and the sample is relatively small, the article has some scientific interest because it corresponds to a specific reality.

However, it needs more clarity in some paragraphs and better systematization in some points. Table 1, for example, mixes several measures to improve water efficiency, which can generally be grouped into three major groups: behavioural measures, technical measures and, eventually, economic measures. In the case of technical measures, they can also be systematized according to the "5R of water efficiency in buildings" principle: reduce consumption (through the use of efficient products), reduce losses and waste, reuse water, recycle water and resort to alternative sources (rainwater, etc.). Authors should try to improve the article in these aspects, where possible.

In the case of rainwater, it is often used for flushing toilets. Was this measure excluded because it was considered difficult to implement in existing buildings? Or for other reasons?

In Figure 3, do the figures above refer to water bill values? It must be clarified.

Author Response

Dear Reviewer 1,

We are grateful to the reviewer for his/her time and suggestions in improving the manuscript. Thank you. 

Reviewer 2 Report

I revised the paper titled “Does the household save water? An evidence from behavioural analysis”. I think that the matter of the paper is in the scope of the Journal and I found it very interesting. The investigation of the consumers’ perception of water-saving and the behavioural usage of domestic water could be important factors to the effort of reducing the water waste. This article provides a comprehensive analysis of the factors which could affect the water consumption behaviour and the perception of water-saving based on the behavior and daily activities of the residents of Rinching Town. However, significant modifications should be done in order to stimulate the overall scientific interest and to make the results more usable and directly comparable to the international bibliography so far.

My suggestions are the following:

Introduction:

The length of the introduction is quite long. Despite, all of this information is very important and interesting, the authors might make an effort to reduce the length (if possible).

Material and methods:

  • Some more information about Rinching Town might be needed. For example, the number of residents, the total area, the drinking water system, the sources/origin of domestic water, etc.
  • Moreover, the description of sample size (line 142-144), the Likert scale description (line 170-174) and the bivariate correlation analysis (line 226-233; 309-313) as well as in what specific analysis they have been used (relationship between factors….., relationship of demographic features….etc), might be referred to this subchapter.

Results:

The statistical analysis of the results and their presence in the manuscript are very satisfactory. However, I am inclined to believe that there is not any further discussion of the results. A simple description of the results that are illustrated in the tables and graphs might not be adequate on its own. Of course, there are a few references to other works but the bibliographic interpretation and the correlation of the results with the findings of other scientific works are necessary. For example, how these results are similar or different from other similar studies, which is the reason for discrepancies, etc.

Conclusion:

Any potential limitations and challenges of this research, as well as any potential proposal for further research, might be referred to in the conclusion

Page 4 – line 141-142: A title for this “demographic” part might be needed.

Page 4 – line 142-144: “The number of………the sample proposal size”. There is a great discrepancy between the proposal sample size (351) and this study sample size (120). How this discrepancy can affect the validity of the results as well as what is the reason that small sample size was used? Moreover, although the small sample size, what exactly have been done to ensure the validity of the results?

Page 4 – line 142-144: It might be removed to Material and methods

Page 5 – Table 2: The percentage for “41 to 50” is 17.5 and not 7.5

Page 5 – line 156: A line space might be needed between table and manuscript

Page 5 – line 170-174: It might be removed to Material and methods

Page 6 – Table 3: A clarification about how Mean (%) are calculated, might be given.

Page 6 – line 180: The 89% of residents have high awareness. Does it make sense? The 89% means that the total number of respondents (120) has an average quite high awareness to save domestic water which is equal to 89% (4.45) of the 5-Point Likert scale. So, how is it possible to consider this number as the percentage of residents with high awareness? The same comment for line 188 (50%), line 373 (83.6%), line 379 (84.6%) and line 380 (79.8%).

Page 7 – Figure 2: The pie charts includes both the number of respondents and the percentage of them. However, some numbers seem as decimal values (for instance 80, 67% in pie chart (b)). Therefore, the symbol of “;” must be used between all the values in all pie charts or another approach could be followed such as “More 3 times; 67% (80)

Page 7 – line 226-233: It might be removed to Material and methods

Page 8– line 239-242: “Table 3……n2=0.042”. Why is there a significant difference since P = 0.293?

Page 8 – line 259: A line space might be needed between table and manuscript

Page 9 – line 261-266: “Table 6……n2=0.039”. Why is there a significant difference in SWLS3 since P = 0.091?

Page 9 – line 268: A line space might be needed between table and manuscript

Page 9 – Table 8: The table includes the confidence interval but there is not any reference to these intervals during the results' discussion. How this interval could be used and what valuable information do they give to us in conjunction with this research?

Page 11 – Figure 3: The title of axis y “Average monthly water bill” might be a little confusing. It seems this axis represents the amount of money and not a percentage. The y axis title might be needed to change.

Author Response

Dear Reviewer 2,

We are grateful to the reviewer for his/her time and suggestions in improving the manuscript.Thank you.

Reviewer 3 Report

Dear Authors

Firstly I would like to say that your article is very interesting from different perspectives, the first is that fresh water is a very scarce resource on the planet and the second perspective is that water is an important factor in ensuring the health of the population.

I believe that this article can help decision makers to create specific measures to make water consumption more rational and reduce it to ensure the sustainability of the supply and the improvement of quality.

From my point of view this article should reinforce the introductory part, I think it is important that they can introduce some of the sustainable development objectives (SDG 17 UN) related to water, as well as talk about some of the negative externalities caused by the misuse of water consumption.

On the other hand, in the conclusions section, I think it would be a good opportunity for this article to introduce some measures that could raise awareness among the different types of consumers so that they can reduce their consumption.

It would also be interesting if in the introduction and conclusions they could make references to technologies that could also improve the reduction of consumption and the reuse of the water consumed.

Below, I would like to share with you some articles where you can obtain ideas, reflections and measures that I believe help to design public policies to improve the management of this important natural resource that is water.

I would like to encourage you to make improvements to the paper so that it can be published. Thank you very much for your efforts.

Proposals:

Keywords;  Evaluation of behavioural, negative externalities, negative externalities water consumption.

Molina-Moreno, V.; Leyva-Díaz, J.C.; Llorens-Montes, F.J.; Cortés-García, F.J. Design of Indicators of Circular Economy as Instruments for the Evaluation of Sustainability and Efficiency in Wastewater from Pig Farming Industry. Water 20179, 653.

Schuetze, T.; Santiago-Fandiño, V. Quantitative assesment of water use efficiency in urban and domestic buildings. Water 20135, 1172–1193

Viles, E.; Santos, J.; Arévalo, T.F.; Tanco, M.; Kalemkerian, F. A New Mindset for Circular Economy Strategies: Case Studies of Circularity in the Use of Water. Sustainability 202012, 9781.

Kayal, B.; Abu-Ghunmi, D.; Abu-Ghunmi, L.; Archenti, A.; Nicolescu, M.; Larkin, C.; Corbet, S. An economic index for measuring firm’s circularity: The case of water industry. J. Behav. Exp. Financ. 201921, 123–129

Argudo-García, J.J.; Molina-Moreno, V.; Leyva-Díaz, J.C. Valorization of sludge from drinking wáter treatment plants. A commitment to circular economy and sustainability. Dyna 2017, 92, 71–75.

Author Response

Dear Reviewer 3,

We are grateful to the reviewer for his/her time and suggestions in improving the manuscript.

Thank you.

Round 2

Reviewer 2 Report

In my opinion, the authors have made an important and severe effort to improve the quality of the manuscript. This version (ver. 2) is much better than the previous (original version) and has covered the majority of the reviewers' comments. However, the discussion of the results might not be still in-depth, although the presentation of the results is quite satisfactory. Also, some other minor suggestions are

Page 2 - line 51-53: "Previous studies......behaviour". The addition of references might be needed.

Line 217, 281, 374, 394, 434, 459: The numbering in the subchapter titles is incorrect.

Author Response

Dear Reviewer 2,   We appreciate the reviewers' time and effort in providing valuable feedback and insightful comments for this paper.   Please see the attachment.   Thank you for allowing us to submit a revised draft of our manuscript.

Reviewer 3 Report

Dear Authors

I have reviewed the article and found that they have incorporated improvements in it. In this sense, from my point of view, it should be published so that it can be useful for future research.

I would like to take this opportunity to wish you a happy end to the year and to encourage you to continue working in this field of research.

Author Response

Dear Reviewer 3,   We appreciate the reviewers' time and effort in providing valuable feedback and insightful comments for this paper.   Please see the attachment.   Thank you for allowing us to submit a revised draft of our manuscript.
